# Cascade electrocatalysis via AgCu single-atom alloy and Ag nanoparticles in $CO_2$ electroreduction toward multicarbon products

Cheng Du[1,8], Joel P. Mills[1,8], Asfaw G. Yohannes [2,8], Wei Wei[1,8], Lei Wang[1], Siyan Lu[1], Jian-Xiang Lian[2], Maoyu Wang[3], Tao Guo [1], Xiyang Wang [1], Hua Zhou [3], Cheng-Jun Sun[3], John Z. Wen [1], Brian Kendall [4], Martin Couillard[5], Hongsheng Guo[5], ZhongChao Tan[1]✉, Samira Siahrostami [2] ✉ & Yimin A. Wu [1,6,7] ✉

Electrocatalytic $CO_2$ reduction into value-added multicarbon products offers a means to close the anthropogenic carbon cycle using renewable electricity. However, the unsatisfactory catalytic selectivity for multicarbon products severely hinders the practical application of this technology. In this paper, we report a cascade AgCu single-atom and nanoparticle electrocatalyst, in which Ag nanoparticles produce CO and AgCu single-atom alloys promote C-C coupling kinetics. As a result, a Faradaic efficiency (FE) of $94 \pm 4\%$ toward multicarbon products is achieved with the as-prepared AgCu single-atom and nanoparticle catalyst under ~720 mA cm$^{-2}$ working current density at −0.65 V in a flow cell with alkaline electrolyte. Density functional theory calculations further demonstrate that the high multicarbon product selectivity results from cooperation between AgCu single-atom alloys and Ag nanoparticles, wherein the Ag single-atom doping of Cu nanoparticles increases the adsorption energy of *CO on Cu sites due to the asymmetric bonding of the Cu atom to the adjacent Ag atom with a compressive strain.

Electrocatalytic reduction of $CO_2$ into valuable chemicals using renewable electricity provides a sustainable route for $CO_2$ recycling and utilization, playing a critical role in realizing a carbon-neutral cycle[1,2]. In the past few decades, enormous progress has been made in single-carbon product generation through electrocatalytic $CO_2$ reduction reaction ($CO_2$RR), especially for carbon monoxide and formic acid[3,4]. However, the application of electrocatalytic $CO_2$RR technology is limited by the low values of the $C_1$ products[5–7]. Therefore, producing multicarbon products from $CO_2$RR seems much more attractive than $C_1$ products[8]. Since Hori et al. reported the production of multicarbon products ($C_2H_4$, $C_2H_5OH$, $CH_3COOH$, n-$C_3H_7OH$, etc.) on copper (Cu) in 1989, Cu has been demonstrated to be the only metal

[1]Department of Mechanical and Mechatronics Engineering, Waterloo Institute for Nanotechnology, Materials Interfaces Foundry, University of Waterloo, Waterloo, Ontario N2L 3G1, Canada. [2]Department of Chemistry, University of Calgary, 2500 University Drive NW, Calgary, Alberta T2N 1N4, Canada. [3]X-Ray Science Division, Argonne National Laboratory, Lemont, IL 60439, USA. [4]Department of Earth and Environmental Sciences, University of Waterloo, Waterloo, Ontario N2L 3G1, Canada. [5]Energy, Mining and Environment Research Center, National Research Council Canada, 1200 Montreal Road, Ottawa, Ontario K1A 0R6, Canada. [6]Interdisciplinary Center on Climate Change, University of Waterloo, Waterloo, Ontario N2L 3G1, Canada. [7]Department of Chemistry, University of Waterloo, Waterloo, Ontario N2L 3G1, Canada. [8]These authors contributed equally: Cheng Du, Joel P. Mills, Asfaw G. Yohannes, Wei Wei. ✉e-mail: zhongchao.tan@uwaterloo.ca; samira.siahrostami@ucalgary.ca; yimin.wu@uwaterloo.ca

that can effectively catalyze $CO_2$ into multicarbon products[9,10]. The sluggish C-C coupling kinetics over pure Cu surface severely hinders the mass production of multicarbon products[11–13]. Accordingly, various catalyst design strategies have been developed to improve the performance of multicarbon products produced using Cu-based catalysts, for instance, alloying, doping, surface modification, and interface engineering[14–17].

Inspired by the complex multistep cascade reactions in enzymes, O'Mara et al. put forward a cascade catalysis strategy to improve the multicarbon production with an Ag@Cu core-shell structured catalyst[18]. The $CO_2$ can be reduced into CO on the Ag core and then transferred to the Cu shell for further reduction into multicarbon products[18]. Similarly, Chen et al. improved the $C_{2+}$ partial current over the Cu catalyst from 37 to 160 mA cm$^{-2}$ by simply adding Ag nanoparticles to the catalyst, proving the effectiveness of cascade catalysis under industrial current densities[19]. Very recently, Zhang et al. applied such a cascade catalysis to the design of gas diffusion electrodes (GDEs)[20]. Specifically, they prepared segmented tandem electrodes, where a CO-selective catalyst layer (CL) segment at the inlet prolongs the CO residence time in the subsequent $C_{2+}$ selective segment, resulting in a high $C_{2+}$ FE over the Cu/Fe-N-C catalysts. These examples evidence the practicability of $C_{2+}$ production by such a cascade $CO_2RR$ mechanism, which integrates two consecutive steps of $CO_2$-to-CO and CO-to-$C_{2+}$ on two distinct catalytic sites. In principle, the C-C kinetics are dominated by two factors, the surface coverage of local adsorbed carbon monoxide (*CO) and the adsorption energy of *CO[12,20,21]. However, the catalyst designs in cascade $CO_2RR$ systems reported earlier only focused on improving the surface coverage of adsorbed *CO by enhancing the local CO concentration, ignoring the adsorption energy of *CO on the Cu surface. Therefore, the $C_{2+}$ performances still cannot satisfy the industrial requirements.

To further break the limitation to $C_{2+}$ performance, we tuned both the surface coverage of adsorbed *CO and its adsorption energy on the C-C coupling site. Specifically, we develop a cascade catalyst where an AgCu single-atom alloy (SAA) serves as a C-C coupling site while the Ag nanoparticles (NP) produce CO locally. The Ag single-atom doping of Cu NP can greatly increase the adsorption energy of *CO on the Cu sites because of the asymmetric bonding of the Cu atom to the adjacent Ag atom with a compressive strain, resulting in a much better C-C coupling ability than that of pure Cu NP. Consequently, the as-prepared AgCu single-atom alloy and Ag NP (denoted as AgCu SANP) catalysts exhibited 94 ± 4% FE toward multicarbon products under the working current density of approximately 720 mA cm$^{-2}$ in a flow cell. In addition, the density functional theory (DFT) calculations indicate the high $C_{2+}$ selectivity mainly is rooted in the cascade catalysis by AgCu SAA and Ag NP.

## Results
### Characterization of the catalysts
The AgCu catalysts were prepared through the galvanic replacement reaction between commercial Cu NPs and Ag$^+$, which was spontaneously driven by their reduction potential difference. As illustrated in Fig. 1A, the dispersion of Ag atoms varies with the increasing amount of Ag$^+$. As a result, AgCu SAA, AgCu SANP, and AgCu NP were synthesized by simply tuning the amount of Ag$^+$. The scanning electron microscope (SEM) images (Supplementary Fig. S1) show the presence of all AgCu catalysts with similar morphology of aggregation with a size between 100 nm to 200 nm. The bright-field scanning transmission electron microscopy (BF-STEM) images (Fig. 1B and Supplementary Fig. S2A, B) further prove the aggregation. The high-angle annular dark-field scanning transmission electron microscopy (HAADF-STEM) images (Fig. 1C and Supplementary Fig. S2C–F) show some small Ag NP in AgCu SANP because of the Z contrast difference between Ag and Cu. At the atomic scale, however, the complex electron scattering between the number of X-rays detected and atoms interacting with the electron probe makes it impractical to directly relate the X-ray counts to the

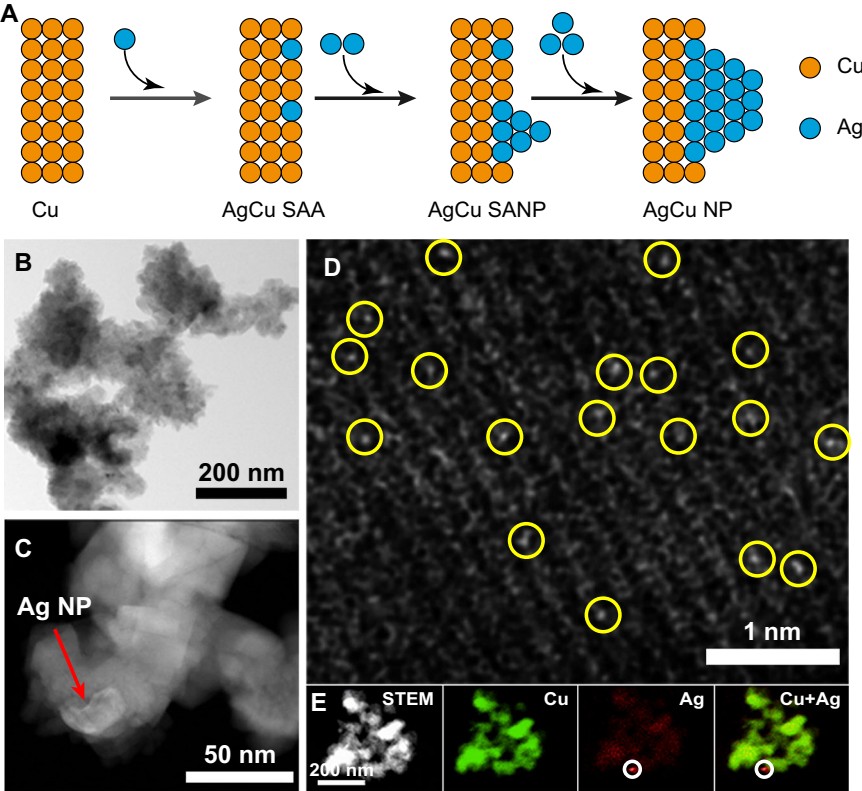

**Fig. 1 | Scanning transmission electron microscopy imaging of catalysts. A** Scheme of the synthesis process of AgCu catalysts; **B** BF-STEM image of AgCu SANP; **C** HAADF-STEM image of AgCu SANP; **D** AC-HAADF-STEM image of AgCu SANP; **E** STEM and EDX element mapping images of AgCu SANP.

number of density of atoms[22,23]. Thus, we cannot confirm the existence of Ag single atoms by the point spectrum (Fig. 2G). Alternatively, atomic scale imaging is required to confirm the existence of Ag single-atoms. The aberration-corrected HAADF-STEM (AC-HAADF-STEM) images (Fig. 1D, Supplementary Fig. S3D) further show many bright dots in the Cu crystal lattice, which are ascribed to the Ag single-atoms. The HAADF-STEM images and the energy-dispersive X-ray spectroscopy (EDX) element mapping evidence the existence of separated Ag NP and the uniform dispersion of Ag and Cu elements in other parts of AgCu SANP (Fig. 1E). These results confirm the co-existence of Ag NP and Ag single-atom in the AgCu SANP. Conversely, the Ag atoms are uniformly and atomically dispersed in the AgCu SAA (Supplementary Fig. S3E).

To further characterize the structure and composition of AgCu SANP, grazing incidence X-ray diffraction (GI-XRD) was measured (Fig. 2A and Supplementary Fig. S4A). The intensity of Cu (111) and CuO (002) and (111) peaks gradually decrease with the increase of Ag in the samples, accompanying the increase of Ag (111) and (200) and $Cu_2O$ (111) and (200) peak intensities. In detail, AgCu SAA shows an XRD spectrum similar to Cu NP, indicating the high atomic dispersion of Ag atoms in AgCu SAA. The co-existence of both Ag and Cu peaks in AgCu SANP confirms the existence of Ag NP, agreeing well with the STEM results. Furthermore, X-ray photoelectron spectroscopy (XPS) was tested to study the elemental valence. Only AgCu SANP and AgCu NP show apparent Ag peaks, which can be deconvoluted into the four peaks of $3d_{3/2}$, $3d_{5/2}$ of metallic Ag, and Ag oxide (as marked in Fig. 2B). The disappearance of the Ag XPS peak in AgCu SAA may result from the low content of Ag, which is only 0.04 wt.% as determined by inductively coupled plasma mass spectrometry (ICP-MS). The core-level Cu $2p$ XPS peak can be deconvoluted into six peaks, corresponding to $2p_{3/2}$, $2p_{1/2}$ of $Cu^0$, $Cu^{2+}$, and shake-up peaks (as marked in Fig. 2C)[24,25]. The Auger spectrum further proves the existence of $Cu^+$ (Supplementary Fig. S4B), and this finding agrees well with the XRD results. In addition, the intensity of Ag and Cu XPS peaks also exhibits a

trend similar to that in the XRD spectra, where Ag increases and Cu decreases with the increasing amount of Ag in the samples. This also supports the reaction mechanism of galvanic replacement. The ICP-MS results show that the exact content of Ag in AgCu SAA, AgCu SANP, and AgCu NP increased from 0.04 wt.%, 1 wt.% to 4 wt.%, respectively.

The coordination structure of the Ag species in as-synthesized AgCu SANP was further analyzed by synchrotron-radiation-based X-ray absorption fine structure (XAFS) at the Advanced Photon Source (APS), Argonne National Laboratory. As shown in the Ag K-edge X-ray absorption near-edge structure (XANES) spectral (Fig. 2D), the adsorption edge ($E_0$) around 0.5 of AgCu SANP shows a slight shift to lower energy compared to that of Ag foil, indicating the electron transfer from Cu to Ag due to the formation of Ag-Cu bond. The positive shift of the adsorption edge of AgCu SANP in the Cu K-edge also demonstrates the electron loss of Cu (Supplementary Fig. S5). Because of the co-existence of $Cu_2O$ and CuO, it is difficult to distinguish the positive shift of the adsorption edge of AgCu SANP in the Cu K-edge is caused by electron transfer from Cu to Ag; or is it for the electron transfer from Cu to O (Supplementary Fig. S5). However, the Fourier transform (FT) of the $k^2$-weighted extended X-ray absorption fine structure (EXAFS) curve of the Ag K-edge of AgCu SANP shows both Ag-Ag and Ag-Cu coordination bonds (Fig. 2E), lacking Ag-O scattering path, which confirms the electron transfer between Cu and Ag. As seen in Fig. 2F, the ratio of Ag-Cu bonds to Ag-Ag bonds was estimated to be around 1:7 based on the extended X-ray absorption fine structure (EXAFS) fitting results (Supplementary Fig. S6 and Supplementary Table S1). These results evidence the co-existence of the single-atom and NP Ag in AgCu. To summarize, a combination of the XANES, the EXAFS analysis, and the AC-STEM images confirms the co-existence of Ag and Cu atoms in the AgCu SAA.

## Performance of $CO_2$ reduction and CO reduction on the catalyst
To demonstrate the concept of cascade catalysis on $C_{2+}$ production (Fig. 3A), the electrocatalytic $CO_2RR$ performance of AgCu SANP and

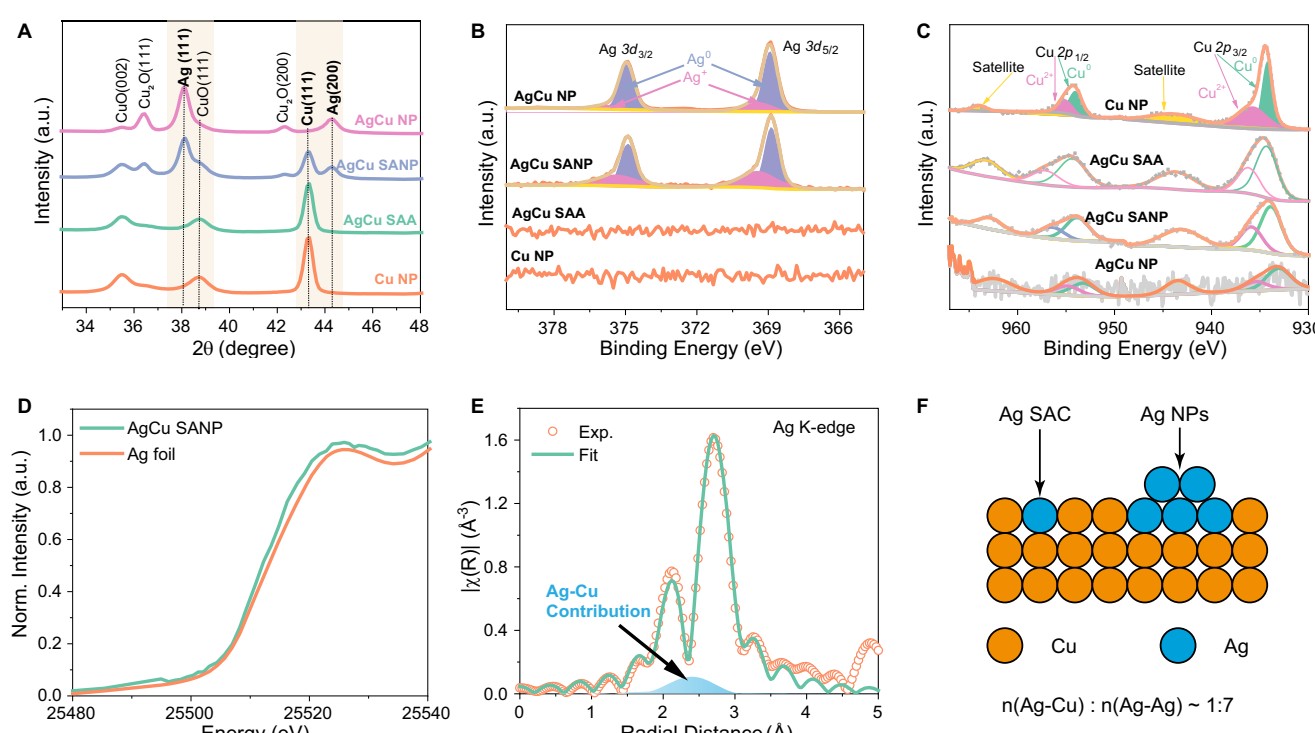

**Fig. 2 | X-ray diffraction and spectroscopies characterization of catalysts.** **A** XRD spectra, **B** Ag 3d core-level XPS spectra, **C** Cu 2p core-level XPS spectra of Cu NP, AgCu SAA, AgCu SANP, and AgCu NP; **D** Ag K-edge XANES spectra of AgCu SANP and Ag foil; **E** Ag K-edge XAFS experimental and fitting spectrum of AgCu SANP; **F** Scheme of AgCu SANP structure and the ratio of Ag-Cu and Ag-Ag contribution from fitting results. Source data are provided as a Source Data file.

other controlled samples were tested in a flow cell with 1 M KOH as an electrolyte; see Supplementary Fig. S7. Specifically, potentiostatic testing was conducted on AgCu catalysts under different potentials. The gas products besides $H_2$ were collected and quantified by in-line gas chromatography (Supplementary Fig. S8), while the liquid products were directly dissolved into the electrolyte and analyzed by nuclear magnetic resonance (NMR) (Supplementary Fig. S9). The $C_{2+}$ products for all catalysts are mainly ethylene and ethanol with minor acetate and n-propanol (Fig. 3B, Supplementary Fig. S10). Typically, AgCu SANP exhibits a $C_{2+}$ FE of $94 \pm 4\%$ under $-0.65$ V, which is much higher than that of Cu NP ($56 \pm 7\%$), AgCu SAA ($78 \pm 3\%$), and AgCu NP ($73 \pm 1\%$) (Fig. 3B). The apparent current density under $-0.65$ V of AgCu SANP ($720 \pm 61$ mA cm$^{-2}$) is also greater than that of Cu NP ($630 \pm 38$ mA cm$^{-2}$), AgCu SAA ($710 \pm 36$ mA cm$^{-2}$) and AgCu NP ($233 \pm 32$ mA cm$^{-2}$) (Fig. 3C). Similarly, Supplementary Fig. S11 shows the $C_{2+}$ partial current density at $-0.65$ V increased from

$353 \pm 21$ mA cm$^{-2}$ (Cu NP), $553 \pm 28$ mA cm$^{-2}$ (AgCu SAA) to $677 \pm 57$ mA cm$^{-2}$ (AgCu SANP), which proves that Ag increases the local CO concentration and expedites the C-C coupling. Too much Ag in AgCu NP causes a rapid decrease of $C_{2+}$ partial current density ($170 \pm 26$ mA cm$^{-2}$) due to the lack of enough Cu sites. In addition, the $C_{2+}$ production performance of as-prepared AgCu SANP is well-placed among the catalysts reported earlier (Fig. 3D). It is worth noting that all the copper oxides in the as-synthesized catalyst were reduced to the metallic state during the negative potential[26]. The FE of CO production is the least in AgCu SANP, which indicates that most of the CO was converted to $C_{2+}$ products via C-C coupling. Again, the concept of the proposed cascade catalysis of AgCu SANP is that the Ag single-atom on Cu promotes the C-C coupling selectivity, while the Ag nanoparticles produce local CO from $CO_2$. To further prove this concept, CO reduction tests were conducted to evaluate the C-C coupling selectivity. As presented in Fig. 3E, AgCu SAA shows a much higher FE than

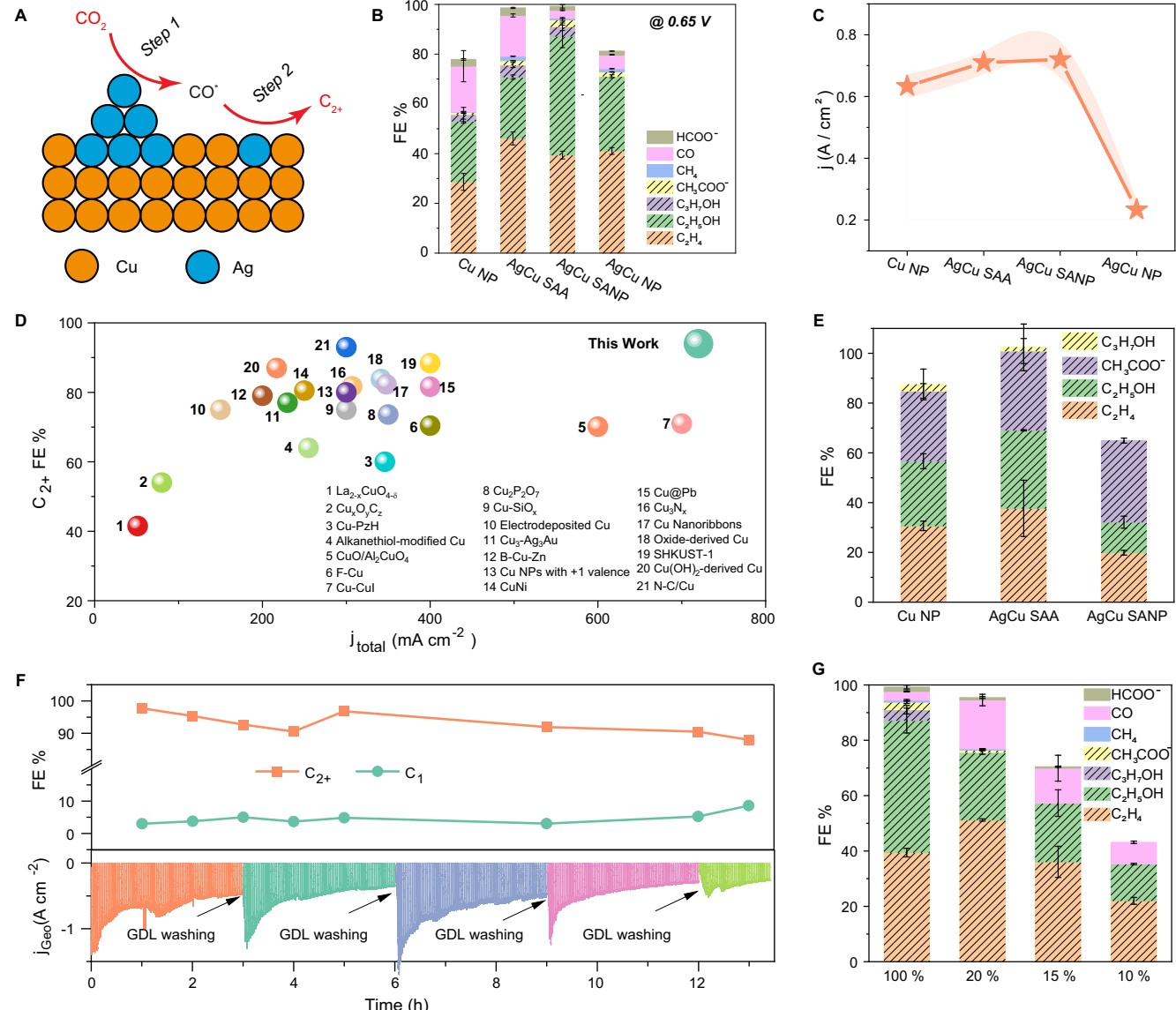

**Fig. 3 | Electrocatalytic $CO_2$ reduction and CO reduction performance of the catalysts. A** Scheme of the cascade catalysis mechanism over AgCu SANP; **B** FE results of Cu NP, AgCu SAA, AgCu SANP, and Ag NP catalysts toward $CO_2$RR at $-0.65$ V; **C** Total current density of Cu NP, AgCu SAA, AgCu SANP, and Ag NP catalysts at $-0.65$ V; **D** Performances comparison of AgCu SANP and reported results; **E** FE results of Cu NP, AgCu SAA, and AgCu SANP toward CO reduction at $-0.65$ V; **F** Long-term stability results of AgCu SANP toward $CO_2$RR; **G** FE results of AgCu SANP toward $CO_2$RR feeding with different $CO_2$ concentrations. Note that the FE of $H_2$ products from competitive hydrogen evolution reaction was not quantified. All the error bars are obtained from three independent experiments. Source data are provided as a Source Data file.

Cu NP does, indicating the strong suppression of HER and the improvement of C-C coupling selectivity by Ag single-atom in Cu lattice. The low FE of AgCu SANP for CO reduction can be ascribed to Ag NP composition, which excludes the adsorption of CO and hinders further CO reduction. This easy desorption of CO from Ag sites is also confirmed by the density functional theory (DFT) calculation (Supplementary Table S3, Supplementary Figs. S14 and S17). Note that the FEs of ethanol and ethylene (green plus orange region) result from $CO_2$ (Fig. 3B) and CO (Fig. 3E) reductions were comparable on AgCu SAA and Cu NP. Conversely, the FEs of ethanol and ethylene (green plus orange region) result from CO reduction on AgCu SANP significant decrease from $87.1 \pm 4.6\%$ (Fig. 3B) to $32.2 \pm 2.4\%$, respectively (Fig. 3E). This is due to the easy desorption of CO from the Ag NP, which decreases the local concentration of CO on the catalyst surface. This also confirms the existence of Ag NP in AgCu SANP as well as the cascade catalysis of AgCu SANP in $CO_2$RR. The FE of acetate formation is similar for CuNP, AgCu SANP, and AgCu SAA in CO reduction (Fig. 3E). This indicates acetate is produced on the Cu active sites that are independent of Ag content. To reduce the flooding in the flow cell, long-term stability was tested by pulse electrolysis (working 90 s and off 30 s) with a polytetrafluoroethylene (PTFE)-based gas diffusion electrode (GDE) as previously reported[27]. Although the current density decreased because of flooding, the $C_{2+}$ FE only slightly decreased from 98% to 88% during the 13-h-long test with a total working time of 9.75 h (Fig. 3F). The catalytic activity of the as-prepared AgCu SANP was also tested using a diluted $CO_2$ gas feed to demonstrate its potential in practical $CO_2$ reduction from industrial flue gases. As shown in Fig. 3G and Supplementary Fig. S12, the $CO_2$RR activity of AgCu SANP remains very high when the concentration of feeding $CO_2$ gas decreased to 20%, exhibiting $95 \pm 2\%$ FE toward $CO_2$ reduction and $76 \pm 1\%$ FE toward $C_{2+}$ production. The $C_{2+}$ FE decreased to $57 \pm 6\%$ (70% for total $CO_2$RR) and $35 \pm 1\%$ (43% for total $CO_2$RR) when the $CO_2$ gas concentration decreased from 15% to 10%, respectively. This result indicates the promising application of AgCu SANP in practical conditions.

## Mechanistic study using density functional theory calculation

Our experiment reveals that ethanol and ethylene are the most common $C_2$ products on the AgCu SAA catalyst with little acetic acid. Both ethanol and ethylene require 12 coupled electron protons, while the formation of acetic acid involves 8 coupled electron protons[28–30]. To model AgCu SAA, we create an Ag-doped Cu structure. To get insight into the atomic-scale $CO_2$ reduction reaction mechanism toward ethylene, ethanol, and acetic acid on the Ag-doped Cu catalyst surface, we performed a detailed mechanistic study considering the possibility of forming different intermediates using DFT calculations. The Cu was confirmed to be polycrystalline in the experiment (Supplementary Fig. S4). However, we model the (100) facet of Cu because it favors $C_2$ products over the $C_1$ products as recommended in the literature[31]. The $CO_2$RR to $C_{2+}$ products involve various intermediates, with the C-C coupling being the rate-determining step[32]. The possible intermediates in each elementary step across the reaction coordinate were examined, as seen in Supplementary Fig. S13. The minimum energy pathways are shown in Fig. 4. Other researchers have reported that the C-C coupling can occur on Cu electrodes through *CO-*CO dimerization, *CO-*CHO, or *CO-*COH with *CO-*CO coupling being less feasible because of a higher transition state (TS) energy (>1 eV)[33–35]. The barriers for hydrogenation of *CO to *CHO and *COH on Cu(100) are 0.64 and 0.94 eV, respectively, which are more favored than the direct CO-CO coupling (>1 eV)[36]. Thus, we choose to investigate the less energetic C-C coupling barriers, i.e., *CO-*CHO and *CO-*COH, on the Ag-doped Cu(100) surface. The Ag doping creates a surface strain because of its larger atomic radius. The Cu-Cu bond length is 2.57 Å on the pure Cu(100) surface, but the Cu-Cu bonds next to the Ag site are compressed to 2.50 Å after doping (Supplementary Fig. S14). The compressive strain creates asymmetrical Cu sites next to the Ag atom (i.e.,

first-neighbor atoms labeled 'B', Supplementary Fig. S15), which provide active sites for the C-C coupling. As shown in Fig. 4A, the TS values of C-C coupling via *CO-*CHO and *CO-*COH reactions are 0.45 and 1.10 eV, respectively. This agrees well with the numbers on Cu(100) reported earlier[33]. To understand the effects of Ag on the reactions, we also calculated the TS barrier of *CO-*CHO on the pure Cu(100) surface. The obtained transition state energy of 0.55 eV barrier is 0.1 eV higher than the Ag-doped Cu(100) surface. On the other hand, Ag is known to produce CO from $CO_2$[37]. The Ag sites, both in the single atom (AgCu SAA) and nanoparticle (AgCu SANP), could convert $CO_2$ to CO. However, the enhanced formation of CO is expected with the latter because of the number of active sites. To assess the formation of CO on a single Ag atom and Ag nanoparticle, we compare the $CO_2$ reduction to CO on the Ag-doped Cu and Ag(100) surface (Supplementary Fig. S16). The result shows that the Ag-doped Cu is a better catalyst than the pure Ag surface in terms of $CO_2$ to CO reduction (Supplementary Fig. S16). However, nanoparticles have more under-coordinated Ag sites (corners and kinks) than the lower Miller index surfaces. Those undercoordinated Ag atoms are the most active for $CO_2$ to CO reduction reaction[38]. The AgCu SANP catalysts are expected to have kinked and corner sites. Therefore, it is likely that the $CO_2$ to CO reduction takes place on both single Ag atoms and Ag nanoparticles.

The CO binding in AgCu SAA is weak on the Ag atom and easily diffuses to the next Cu atoms (Supplementary Table S3 and Supplementary Fig. S17). These findings suggest that the Ag atoms play the following two key roles. (1) facilitating the reduction of $CO_2$ to CO and (2) expediting the C-C coupling by spilling over the CO atom to this asymmetrical active site with a reduced transition state energy.

## Ethanol, ethylene, and acetic acid pathways

The lowest energy pathways identified for the production of ethanol, ethylene, and acetic acid are illustrated in Fig. 4B, C. All the intermediates before the C-C coupling are the same for these three products, i.e., $CO_2 \rightarrow$ *COOH $\rightarrow$ *CO $\rightarrow$ *CHO $\rightarrow$ *CO-CHO. Because of the lower TS barrier, all the elementary steps after C–C coupling are investigated based on the *CO-CHO intermediate. The acetic acid path bifurcates at the sixth coupled proton-electron step, while ethanol and ethylene share six-coupled proton-electron transfer and bifurcate only at the seventh reduction step. The common intermediate for ethanol and ethylene, i.e., *OCH-CHO* (labeled **7A**), is more favored than the acetic acid intermediate (labeled **7B**), indicating the formation of ethanol and ethylene before acetic acid (Fig. 4C). This mechanistic understanding supports the experimental finding in Fig. 3B, where ethylene and ethanol are the major products, and a negligible amount of acetate/acetic acid is formed in $CO_2$RR. However, the formation of acetate from CO electroreduction reaction (CORR) is equivalent to the formation of ethylene and ethanol (Fig. 3E). This can be due to the presence of a high amount of CO on the catalyst surface that expedites the C-C coupling step and thereby enables the formation of other $C_2$ products including the acetate/acetic acid. As shown in Fig. 4, the free energy of the acetic acid formation step (labeling **7B**) is slightly higher (0.19 eV) than those of ethanol and ethylene (labeling **7A**). Therefore, the acetic acid/acetate pathway would be feasible as well.

Comparing ethanol and ethylene formation from $CO_2$ (which both are 12-electron reduction products), free energy diagram analysis revealed that the uphill steps after the coupling for ethanol and ethylene ($CH_2CH_2OH^* \rightarrow CH_3CH_2OH$ and $CH_2CHO^* \rightarrow CH_2CH_2O$) are 0.4 and 0.44 eV, respectively. This negligible difference of 0.04 eV is within the DFT error, which indicates that both ethanol and ethylene are equally probable products of $CO_2$RR on Ag-doped Cu. Nonetheless, the solvent, pH, and double-layer electric field effect were excluded from the DFT calculations, which could affect the stabilization of different intermediates and the consequent selectivity of the final products[39,40].

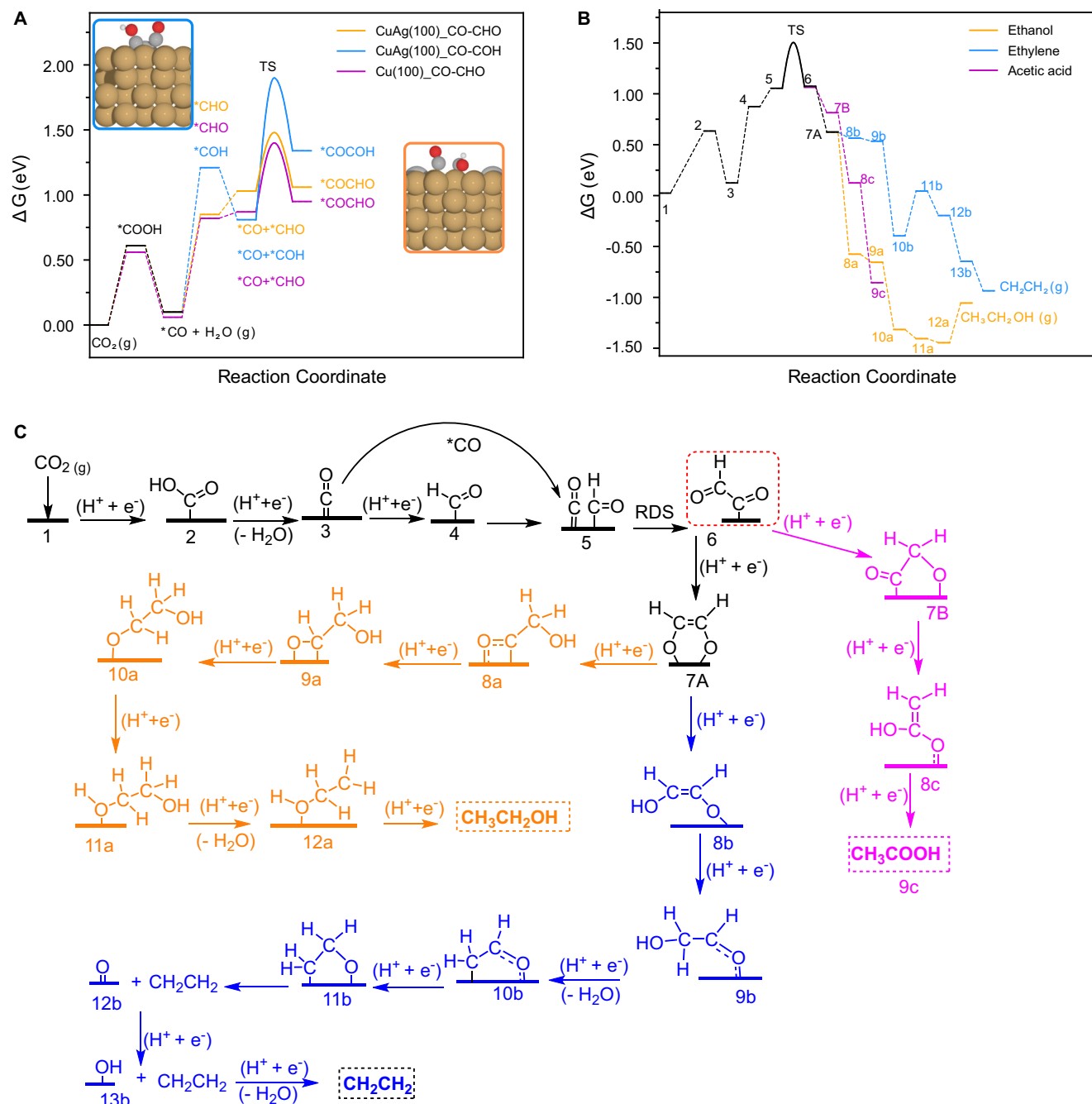

**Fig. 4 | Mechanistic studies by the density functional theory calculations.** Mechanistic studies by the density functional theory calculations. **A** Comparison of the C-C coupling activation barrier for the *CHO (on CuAg and pure Cu) and *COH intermediates (CuAg); **B** The lowest free energy pathway for the formation of ethanol (orange), ethylene (blue) and acetic acid (magenta) on CuAg and **C** the associated chemical formula for each elementary step in (**B**). Source data are provided as a Source Data file.

Comparing ethanol and ethylene (which both are 12-electron reduction products), free energy diagram analysis demonstrates that ethanol formation is favored over ethylene (labeling **8a** and **8b** in Fig. 4B, respectively). This agrees with the slightly higher percentage of ethanol over ethylene in the experimental result for the AgCu SANP shown in Fig. 3B.

To gain further insight into the impact of Ag doping on the electronic structure of copper surfaces, we analyzed the density of states (DOS) of the pure Cu and Ag-doped Cu surfaces (in the absence of adsorbate). Supplementary Fig. S15 shows the $d$-band, which has shown to be a good descriptor for understanding the differences in catalytic activities[41]. This analysis shows that the $d$-band center in pure

Cu(100) surface is shifted to lower values in the Ag-doped Cu, as indicated by the vertical dashed lines. This downshift originates from the excess of DOS in Ag-doped Cu (compared to pure Cu) concerning the Fermi level. The projected DOS on the $d$ orbitals of the Cu (B) atoms near the Cu active site and Ag atom (before and after Ag substitution, respectively) also shows a downshift of the $d$-band center upon Ag substitution. The downshift of the $d$-band center in the Ag-doped Cu surface indicates that Ag reduces the bonding strength of the adsorbate at the surface, which is consistent with the weak adsorption of CO on the Ag-doped Cu surface, and the reduced reaction barriers to producing ethanol, ethylene, and acetic acid molecules (Fig. 4A, B).

## Discussion

In conclusion, an AgCu SANP cascade catalyst has been developed and exhibited an FE of 94 ± 4% toward multicarbon products under a working current density of 720 mA cm$^{-2}$ or so in a flow cell. DFT calculations indicate that the high $C_{2+}$ selectivity is ascribed to the synergistic catalysis of Ag NP and AgCu SAA in the AgCu SANP, namely Ag NP generating CO and AgCu SAA promoting the C-C coupling step. This work not only reports a highly effective catalyst for multicarbon product production but also provides a cascade catalysis strategy for future $C_{2+}$ selective catalyst development.

## Methods

### Chemicals

All solvents, unless stated otherwise, are of ACS grade. Copper nanoparticles with an average diameter of 100 nm and silver nitrate (AgNO$_3$, 99%) were purchased from Sigma-Aldrich, Canada. The water used was purified by a Milli-Q system with a resistivity of 18.2 MΩ cm. All standard gases for electrocatalysis and gas chromatography were obtained from Praxair and Linde, Canada, with the associated purities: CO$_2$ LaserStar 5.0 (99.999%), Air Zero (maximum 3ppm water, 1 ppm total hydrocarbons), Helium 5.5 (99.9995%), Hydrogen 5.5 (99.9995%), and Argon 5.0 (99.9995%). The gas mixture for GC calibration (Supelco 23462) was obtained from Sigma-Aldrich too. For the NMR tests, deuterium oxide (99.9 atom % D), dimethyl sulfoxide (99.5%), formic acid (98%), and 1-propanol (99.9%) were also purchased from Sigma-Aldrich. Ethylene glycol (laboratory grade) was obtained from VWR Chemicals. Nafion 117 containing solution (~5% in a mixture of lower aliphatic alcohols and water) was also obtained from Sigma-Aldrich. Sigracet 36 BB gas diffusion electrodes were purchased from the Fuel Cell Store. Nickel foam (99.9%) was purchased from MTI Corporation. Potassium hydroxide (85%) was obtained from Sigma-Aldrich.

### Synthesis of AgCu SANP

First, 20 mg Cu NPs and 2 mg AgNO$_3$ were dispersed into 5 mL of ethylene glycol and 0.2 mL of water, respectively. These two solutions were then mixed and put into an ultrasonic bath for the galvanic reaction for 30 min. After that, the nanoparticles were washed with water and isopropanol and collected using a centrifuge. The obtained nanoparticles contain both Ag single-atoms and nanoparticles labeled as AgCu SANP. The AgCu SAA and AgCu NP were prepared following a similar procedure, except for the different doses of AgNO$_3$ (0.1 mg for AgCu SAA and 20 mg for AgCu NP). The ethylene glycol solvent is reductive, and the ultrasonic bath inevitably raises the temperature of the solvent. Thus, the ethylene glycol can partly reduce the CuO in the Cu NPs into Cu$_2$O (Fig. 2A).

### Preparation of GDE

AgCu-based nanoparticles were first dispersed into IPA with a concentration of 5 mg mL$^{-1}$. The Nafion-117 solution was also added to the IPA solution with a 1% volumetric ratio. After mixing with ultrasound for 30 min, the catalyst ink was directly sprayed onto the GDL (Sigracet 36 BB) using an airbrush. The final loading of GDE was determined by balancing the weight difference before and after the airbrush step, which was about 0.5 mg cm$^{-2}$.

### Material characterization

Scanning electron microscopy (SEM) images were captured on a Hitachi S4800 with a working accelerating voltage of 10 kV. Glancing-incidence X-ray diffraction (GI-XRD) was measured on a PANalytical X'Pert Pro MRD diffractometer with Cu Kα radiation (1.54 Å) at an incidence angle of 0.3°. X-ray photoelectron spectroscopy (XPS) measurements were carried out on a Thermo-VG Scientific ESCALab 250 microprobes with a monochromatic Al Kα X-ray source (1486.6 eV). The obtained spectra were calibrated using the C 1s line. Aberration-corrected scanning transmission electron microscope (AC-

STEM) tests were carried out on an FEI Titan 80–300 HB TEM equipped with energy-dispersive X-ray spectroscopy (EDX) at 200 kV. The HAADF-STEM images were recorded by FEI Titan 80–300 HB TEM/STEM with an aberration corrector operating at 300 kV. Inductively coupled plasma mass spectrometry (ICP-MS) analyses were carried out on an Agilent 8800 triple quadrupole, using He as a collision cell gas, Ge and In as internal standards to correct for instrument drift and ICP element standards (secondary standards from Delta Scientific Inorganic Ventures; primary calibration standards from Alfa Aesar and Aristar VWR Chemicals BDH) to confirm instrument accuracy (within 3%; relative standard deviation for individual sample analyses was ≤12%). Proton nuclear magnetic resonance (H-NMR) was measured on Bruker Avance III 300 MHz. GC tests were conducted on an Agilent 6890 machine with Carboxen (TCD) and Carbonplot (FID) columns. XAS measurements were carried out at the Advanced Photon Sources (APS) 20-ID-C and 20-BM beamlines. The measurements at the copper K-edge and Ag K-edge were performed in fluorescence mode using a Lytle detector. The XAS data were analyzed using the software package Athena.

### Electrochemical CO$_2$RR and CORR tests

All electrochemical tests were measured on Gamry Reference 3000 electrochemical workstation at room temperature (23 ± 2 °C) with IR compensation. A three-electrode system was fabricated with the prepared AgCu-based GDE, Ni foam, and Ag/AgCl electrode serving as working electrodes, the counter electrode, and the reference electrode, respectively[42]. A gas-tight, three-chamber flow cell equipped with a piece of Fumasep FAB-PK-130 anion exchange membrane (AEM) was employed to conduct the electrochemical reaction. Before CO$_2$RR tests, the AEM was first activated in 1 M KOH solution for 12 h. The CO$_2$RR catalytic activities were evaluated using the potentiostatic technique under selective potential for 10 min in 1 M KOH with flowing pure CO$_2$ gas. The flow rate of electrolyte and CO$_2$ gas was 18 mL min$^{-1}$ and 30 sccm, respectively. The gas flow was measured by a flow meter at the end of the GC output tube to keep it the same as the flow rate in the standard curve test. All the potential values are relative to the reversible hydrogen electrode (RHE) and follow the equation below unless stated otherwise.

$$E(\text{RHE}) = E(\text{Ag/AgCl}) + 0.059 * \text{pH} + 0.197 \qquad (1)$$

The IR compensation was conducted automatically by the Gamry Reference 3000 electrochemical working station. After opening "Chronoamperometry" in the Gamry measurement software, parameters such as applied voltage, test duration and IR compensation mode can be configured. The Current Interrupt (CI) mode of the potentiostat (Gamry Reference 3000) automatically compensates for internal resistance and adjusts compensation values in real time based on changes in internal resistance. The electrolyte for both the cathode and anode is 30 mL of Ar-saturated KOH solution separately. The gas products were tested using online GC, and the liquid products were detected using H-NMR. For CORR tests, the feeding gas was replaced with CO, other and conditions are the same as those for CO$_2$RR tests. Three independent experiments were carried out to obtain all error bars.

### Determination of gas products using GC

The standard curves of CH$_4$, CO, C$_2$H$_2$, C$_2$H$_4$, and C$_2$H$_6$ gases were first built using corresponding standard gases. Different concentrations were prepared by diluting the mixture with CO$_2$ using mass flow controllers (Alicat Scientific). The produced standard curves are shown in Supplementary Fig. S8. During the electrocatalytic CO$_2$ reduction, the gas products flow into the GC with the input CO$_2$ gas in the online tube. CH$_4$ and C$_2$H$_4$ are the main gas products that can be detected in the flame ionization detector (FID), and CO was detected by the thermal

conductivity detector (TCD). There is no $CO_2$ reduction gas product being detected in the GC from the anode. $H_2$ was not quantified in this work due to the inability of the GC.

## Determination of liquid products using the NMR
The standard curves of ethanol, formate, acetate and propanol were first built using pure chemicals with known concentrations. NMR tubes were prepared by combining 630 μL of sample with 70 μL $D_2O$ and 30 μL of aqueous 5 mM DMSO internal standard. The produced standard curves are shown in Supplementary Fig. S8. After electrocatalytic $CO_2$ reduction, the electrolyte was collected and mixed with the NMR detecting solution (DMSO internal standard DMSO, D2O, as above). For H-NMR tests, 128 scans were performed, with excitation sculpting used to suppress the water peak. As seen in Supplementary Fig. S9, ethanol, formate, acetate, propanol, and DMSO can be ascribed to the peaks located at 1.07 (triplet) and 3.55 (quartet), 8.35 (singlet), 3.44 (triplet), and 1.42 (sextet) and 0.77 (triplet), and 2.61 ppm, respectively. The signals from the OH proton in the alcohols are not observed, likely because of hydrogen-deuterium exchange. There is no $CO_2$ reduction liquid product being detected in the NMR from anolyte.

## FE calculations
FE represents the ratio between the electrons used for product generation and the total electrons during the reaction[43], namely:

$$FE = \frac{Amount\ of\ the\ product \times n \times F}{\int_0^t i\,dt} \quad (2)$$

where $n$ is the number of moles of electrons to participate in the Faradaic reaction; $t$ is the time of electrolysis; $F$ is the Faraday constant, i.e., 96485 C mol$^{-1}$. The reported values were calculated based on three separate measurements under the same conditions.

## DFT calculations
All DFT calculations have been performed using the Vienna ab initio simulation package (VASP) with spin polarization[44]. The generalized gradient approximation (GGA) exchange-correlation functional parametrized by Perdew, Burke, and Ernzerhof (PBE) for the electronic interactions, and the projector-augmented wave (PAW) method for the core electrons was used[45,46]. A cutoff energy of 600 eV was used for the plane wave. The convergence criterion for the electronic self-consistent iteration was set at $10^{-6}$ eV. During the relaxation, we assumed the relaxation was achieved when the atomic forces were lower than 0.05 eV/Å. We used a Monkhorst–Pack grid with dimensions of $4 \times 4 \times 1$ for sampling the first Brillouin zones[47]. The DOS calculations were done using a $12 \times 12 \times 12$ $k$-points grid and the tetrahedron method with Blöchl corrections[48]. The $d$-band center was calculated with an integration window of −10 to 10 eV. The copper surface was modeled using $2 \times 2$ supercells of the fcc(100) with four layers, for which only the bottom two layers were frozen while the rest of the system was allowed to relax. The Cu(100) is doped with Ag by substituting one Cu atom with Ag, which corresponds to a 3% Ag doping concentration on the Ag-doped Cu surface. To ensure that the interactions between neighboring periodic images are negligible, a vacuum region along the z-direction has been added so that the distance between the two nearest surface atoms in neighboring images is at least 16 Å. We employed the DFT-D3 Grimme method for long-range dispersion interaction correction[49]. For the C-C coupling, a transition state search was carried out by automated relaxed potential energy surface scans (ARPESS)[50]. The computational hydrogen electrode (CHE) model proposed by Nørskov et al. was used to calculate the free energies of $CO_2$ reduction intermediates, based on which the free

energy of an adsorbed species is defined as:

$$\Delta G_{ads} = \Delta E_{ads} + \Delta E_{ZPE} - T\Delta S_{ads} + \int C_P dT \quad (3)$$

where $\Delta E_{ads}$ is the electronic adsorption energy, $\Delta E_{ZPE}$ and $T\Delta S_{ads}$ represent zero-point energy and entropy (difference between adsorbed and gaseous species), respectively, $\int C_P dT$ is the enthalpy correction and $T$ is at room temperature[51].

## Data availability
The authors declare that all data supporting this study are available within the paper and Supplementary Information files. Source data are provided with this paper.

## Code availability
All the DFT calculations were performed using the commercial software VASP. All the input and output files of the calculations are available per request.

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

## Acknowledgements

Y.A.W. thanks the funding from the Tang Family Chair in New Energy Materials and Sustainability, National Research Council of Canada Materials for Clean Fuels Challenge Program (CH-MCF-123), the Natural Sciences and Engineering Research Council of Canada (NSERC) (RGPIN-2020-05903, GECR-2020-00476), New Frontiers Research Fund-Transformation (NFRFT-2022-00197), Canadian Foundation for Innovation John R. Evans Leaders Fund ((#41779), and Ontario Research Fund for Small Infrastructure Fund ((#41779). S.S. and A.Y. acknowledge the support received from the National Research Council of Canada's Materials for Clean Fuels (MCF) Challenge R&D Program (Grant number CH-MCF-115-1). This research was made possible in part thanks to the computational resources provided by Compute Canada (www.computecanada. ca). J.X.L. acknowledges the support from the University of Calgary's Canada First Research Excellence Fund Program, the Global Research Initiative in Sustainable Low Carbon Unconventional Resources. This research used resources of the Advanced Photon Source, an Office of Science User Facility operated for the U.S. Department of Energy (DOE)

Office of Science by Argonne National Laboratory and was supported by the U.S. DOE under Contract No. DE-AC02-06CH11357, and the Canadian Light Source and its funding partners.

## Author contributions

Y.A.W. conceived and supervised the project. S.S. led the DFT calculations. C.D. and J.P.M. carried out the catalyst's synthesis, characterization, and performance test. A.G.Y. and J.X.L. conducted the DFT calculations. M.W., H.Z., and C.J.S. carried out the XAFS measurements and data fitting. S.L., W.W., and L.W. help with electrochemical measurements. T.G. and X.W. assisted in the material characterizations. J.Z.W. and H.G. provided advice for the research and the analysis of results. B.K. performed ICP test. M.C. measured the STEM experiments. C.D., J.P.M., A.G.Y., Z.C.T., S.S., and Y.A.W. wrote the manuscript. All authors made comments and revised the manuscript.

## Competing interests

The authors declare the following competing interests: Y.A.W., C.D., J.P.M., and H.G. have filed a patent application through the University of Waterloo on this technology related to this CuAg single atom alloy for the electrocatalytic $CO_2$ reduction processes (US Patent application number 63/473,924). The remaining authors declare no competing interests.
