## [Peer Review File · Nature Communications]

REVIEWER COMMENTS

Reviewer #1 (Remarks to the Author):

This work reported AgCu single-atom alloy catalysts prepared by the galvanic replacement reaction between commercial Cu NPs and AgNO₃. The catalyst showed outstanding Faradaic efficiency of C₂⁺ products (94%) and high current density (720 mA cm⁻²) in CO₂ reduction reaction. The performance of this catalyst is very attractive and may qualify this work to be published on Nature Communication. However, the characterization data of the catalysts were poorly analyzed and the DFT simulation result is not very convictive. Therefore, I recommend this work not to be accepted in the present form. The following comments need to be addressed.

1. In the XRD patterns, as the galvanic replacement proceeded, why the diffractions of CuO became lower while the diffractions of Cu₂O became stronger?
2. Line 79-Line 81: "One should note that it contains Cu (111), (200), and (220) reflections in all the samples (Figure S4), which indicates Cu is polycrystalline." What does "polycrystalline" here mean? Does it mean the whole sample was polycrystalline or a Cu nanoparticle is polycrystalline? A powder sample of single-crystalline Cu nanoparticles will also generate diffractions of different facets since the nanoparticles are not oriented through any self-assemble process.
3. XRD patterns indicate the existence of Cu₂O, but Cu⁺ is not considered in the XPS fitting in Figure 2C. Usually, large satellite features indicate the existence of Cu²⁺. Cu⁺ doesn't show strong satellite feature. The binding energies of 2p electron in Cu⁺ and Cu⁰ are similar, but Cu⁺ and Cu⁰ can be distinguished by Auger spectrum. Therefore, the Auger spectrum of the catalysts should also be analyzed.
4. Figure S2G: "EDX of region 2, which only shows the signal of Cu without Ag due to the low content of Ag in AgCu SAA composition." If Ag single atoms can not be detected by EDX, then why the EDX mapping in Figure S3E still shows the uniform distribution of Ag? An enlarged EDX spectrum of the AgCu SAA region need to be provided to show whether the signal of Ag is above the noise.
5. In the EXAFS fitting of AgCu SANP, the Ag-Ag and Ag-Cu coordination number are 7 and 1, respectively. Indicating more Ag atoms were in Ag NPs instead of SAA. The Cu-Cu (metal + oxide) and Cu-Ag coordination number are 3.5 and 1, respectively. Considering that the fraction of Ag was only 1 wt% in this sample and only a small fraction of Ag atoms was SAA, the ratio of coordination numbers of Cu-Ag/Cu-Cu as 1:3.5 seems not very reasonable. The coordination number of Cu-Ag seems too high.
6. Line 97-98: "The positive shift of the adsorption edge of AgCu SANP in the Cu K-edge also demonstrates the electron loss of Cu (Figure S5)." The XRD has shown the existence of CuO and Cu₂O in this sample. The edge positions of CuO and Cu₂O are more positive than Cu foil. Therefore, the positive shift of the edge did not indicate the electron transfer from Cu to Ag. The standard spectra of CuO and Cu₂O should be added in Figure S5A. If possible, the author should reduce the sample by H₂ at relative

low temperature to remove the oxides and keep the Cu-Ag distribution, then collect the XAS spectra, to exclude the interference from oxide.

7. In the EXAFS fitting of Cu in AgCu SANP, the bond length of first Cu-O shell was 1.952 Å, longer than the Cu-O bond in CuO (1.947 Å) and Cu₂O (1.849). What is the possible reason? The existence of Cu₂O was not considered in the fitting.

8. Line 125-126, "As presented in Figure 3E, AgCu SAA presents a much higher FE than that of Cu NP, strongly indicating the improvement of C-C coupling kinetics by Ag single-atom in Cu lattice." The FE of Cu NP is about 87%, and that of AgCu SAA is around 100%. This only indicate HER was suppressed on AgCu SAA. To show the kinetics on two catalysts, you should compare the reaction rate, namely partial current density, rather than FE of C₂⁺ products. Similarly, plots of partial current density should be provided for CO₂ reduction. Since the total current densities on Cu and AgCu ASNP are similar (Figure 3C), the enhancement factor of partial current density of C₂⁺ on AgCu ASNP seems less than 2 compared with Cu. The difference between AgCu SAA and AgCu ASNP is even smaller. The enhancement factor less than 2 is not very significant in kinetics. The increasing of FE of C₂⁺ is more related to the suppression of formation of other minor products.

9. Following last comment, to compare the intrinsic kinetics of C₂⁺ formation on different catalysts, maybe its better to conduct the CO₂ reduction tests in H cell with low loading of catalyst, to guarantee all catalyst particles are accessible for CO₂ in solution. The current density needs to be kept low to avoid the rate being limited by mass transport. Then by comparing the ECSA normalized partial current density of C₂⁺ products, the intrinsic activity of different catalysts can be compared. Under this situation, the FE of C₂⁺ may be much lower than the value in flow cell, but more kinetic insights can be obtained.

10. Line 152-153, "To model AgCu SAA, we adapt an Ag-doped Cu model structure." What is the structure of this "Ag-doped Cu model"? Line 167-168, "In pure Cu(100) surface the Cu-Cu bond length is 2.57 Å, however, after doping, this is compressed to 2.50 Å at the surface." Which Cu-Cu bond is compressed? The bond between two Cu next to Ag atom? Please draw the structure of cells used for simulation and indicate the bond mentioned.

11. Line 161-163: "Prior studies have reported that on Cu electrodes, the C-C coupling can occur through *CO-*CO dimerization, *CO--*CHO, or *CO-*COH with *CO-*CO coupling being less feasible with a higher transition state (TS) energy (> 1 eV).³¹⁻³³" Do the authors mean *CO-*CHO and *CO-*COH couplings are less feasible than *CO-*CO coupling? If so, why the author only simulated *CO-*CHO and *CO-*COH couplings in the following sections. I think more previous works regard *CO-*CO coupling as the predominant pathway for C-C bond formation, including references 31-33. In some reports, *CO-*CO is an electron-transfer step decoupled with proton transfer. Decoupled electron-proton transfer is more difficult to treat in DFT simulation. Is this part of the reason the authors chose to simulate *CO--*CHO and *CO-*COH couplings.

12. Figure 4a and 4b only show the energy diagram on Ag-doped Cu (100). The authors should overlay the energy diagram on pure Cu(100) to show how much difference the Ag atoms induced. Line 170-174: "As shown in Figure 4A, the TS values of C-C coupling via *CO-*CHO and *CO-*COH reactions are 0.51 and 1.10 eV, respectively, ... A transition state energy of 0.55 eV barrier is obtained, which is slightly higher than the Ag-doped Cu(100) surface." What is the common error for DFT simulated activation

energy? Is 40 meV difference of couple barrier a significant difference? Does the doping of Ag significantly facilitate the coupling step?

13. Line 174-175: "The main difference between the selectivity of Cu(100) and Ag-doped Cu(100) thus comes from the fact that Ag is deemed to produce CO from CO₂." Does "Ag" here mean Ag atom or Ag particle? Is Ag single-atom in Cu or Ag NP the catalyst for CO₂-to-CO? If Ag single-atom can play this role, then why AgCu SANP showed higher selectivity to C₂+ than AgCu SAA in CO₂ reduction. Maybe, the authors should simulate the energy diagram of CO₂-to-CO on Ag single-atom and Ag metal to show which one is more active for this step and whether Ag NPs are necessary for the tandem reaction.

14. Line 211: The Chemical section did not provide the information of Cu NPs and AgNO₃, the most two important chemicals used in this work.

Reviewer #2 (Remarks to the Author):

The author designed an AgCu single-atom alloy and Ag nanoparticle cascade catalyst for CO₂ to C₂+ products conversion. The catalyst demonstrates 94% total C₂+ products selectivity under 720 mA cm⁻² current density. The performance ranks at the top level of all the published results. Moreover, the author used a simple and reproducible synthesis method. In the meanwhile, the characterization is comprehensive and detailed. Overall, I support the publication of this manuscript if the following concerns are addressed.

When the AgCu SANP significantly improves the C₂+ FE compared to Cu NP and AgCu SAA, the current density remains in a similar range. If the Ag increases the local CO concentration and expedites the C-C coupling, the improvement in current density should be more obvious. Could the author explain more about that?

Could the author explain why there is a 20% missing for the total FE of Cu NP in Figure 3B? It is not a fair comparison to say AgCu SAA improves C₂+ FE if the Cu NP result has a large error. Similarly, the AgCu NP data in Figure 3B and AgCu SANP data in Figure 3E all miss a significant part of the total FE.

Figure 3F uses "second" as the unit for the time scale, which is not reading-friendly. The author should use "hour" as a unit.

The author claims that ethanol formation is favored over ethylene. Then, why do Cu NP and AgCu SAA generate more ethylene than ethanol during CO₂RR?

Reviewer #3 (Remarks to the Author):

In this paper, the authors disclosed the synergy effect between AgCu SAA and Ag nano particles for efficient CO₂RR. Although a high FE of 94% C₂+ products was achieved on their AgCu SANP, no enough evidence could support their results and mechanism.

1. Although a special structure was claimed, the traditional tandem reaction mechanism was still used to explain their results. How is the meaning to constructure this new structure?
2. With the increase of the Ag, there is no doubt to constructure Ag nanoparticles. How about the proportion of Ag SA and nanoparticles? What is the main contribution to the reaction? How to determine Ag SA is close to the Ag nanoparticles? How many Ag SA will improve the C-C coupling and what is the case of nanoparticles close to each other? What is the effect of Ag SA far from the Ag nanoparticles?
3. Usually, Ag shows weak adsorption to *CO. Why the authors considered it will increase the adsorption of *CO for Cu? It is wired that the adsorption of *CO could be enhanced by a weak adsorption one.
4. From the CO₂RR, the authors just take the very beginning 100s or less data. In the early stage of the reaction, there are always some fake data. Thus, data from over 20 min reaction is recommended to show its performance.
5. In the CO reduction, the AgCu SANP showed much lower ethanol generation than that of AgCu SAA and Cu NPs. While, it showed much higher ethanol generation than those of AgCu SAA and Cu NPs in CO₂RR. This is opposite for their explanation. The same problems are similar for ethylene and acetic acid.
6. The DFT simulation is too simple.
7. There many typing error, such as Figure S7, S8 were wrong in the experimental part.

Point-to-Point Response

Reviewer #1 (Remarks to the Author):

This work reported AgCu single-atom alloy catalysts prepared by the galvanic replacement reaction between commercial Cu NPs and AgNO₃. The catalyst showed outstanding Faradaic efficiency of C₂⁺ products (94%) and high current density (720 mA cm⁻²) in CO₂ reduction reaction. The performance of this catalyst is very attractive and may qualify this work to be published on Nature Communication. However, the characterization data of the catalysts were poorly analyzed and the DFT simulation result is not very convictive. Therefore, I recommend this work not to be accepted in the present form. The following comments need to be addressed.

1. In the XRD patterns, as the galvanic replacement proceeded, why the diffractions of CuO became lower while the diffractions of Cu₂O became stronger?

Response:

The synthesis of AgCu catalyst is mentioned in the methods section in page 9 of the manuscript: “20 mg Cu NPs and 2 mg AgNO₃ were firstly dispersed into 5mL ethylene glycol and 0.2mL H₂O, respectively. Then, the two solutions were combined being together and put in an ultrasonic bath for the galvanic reaction for 30 min.” The ethylene glycol solvent is reductive, and ultrasonic bath unavoidably increased the temperature of solvent. So, the ethylene glycol can partly reduce the CuO in the Cu NPs into Cu₂O. That’s the reason why diffractions of CuO became lower while the diffractions of Cu₂O became stronger. We add this explanation in the method section in page 9 of the manuscript. It states “The ethylene glycol solven is reductive, and the ultrasonic bath unavoidably increased the temperature of solvent. Thus, the ethylene glycol can partly reduce the CuO in the Cu NPs into Cu₂O (Figure 2a).”

To better understand the synthesis method, we want to explain a little more about the method design. The AgCu catalysts were prepared through the galvanic replacement reaction between commercial Cu NPs and Ag⁺, which was spontaneously driven by their reduction potential difference. (Advanced Materials, 2013, 25(44): 6313-6333.) However, this replacement reaction is very rapidly due to the relative strong oxidation of Ag⁺, and easy to form atoms aggregation rather than single-atom catalysts. Hence, we utilize reductive ethylene glycol solvent to slow down the replacement reaction between Ag⁺ and Cu to control the resulted catalysts. Small amount of water (0.2mL) was added to dissolve the AgNO₃ salt since it is hard to dissolve in ethylene glycol. Ultrasonic bath was used to increase the dispersibility of Cu NPs in the reaction solvent, but the solvent temperature increase was unavoidable. Therefore, the CuO in the Cu NPs were partly reduced into Cu₂O by the ethylene glycol under high temperature. By the way, the Cu oxidized by Ag⁺ was in the form of Cu²⁺ rather than Cu oxidation.

2. Line 79-Line 81: “One should note that it contains Cu (111), (200), and (220) reflections in all the samples (Figure S4), which indicates Cu is polycrystalline.” What does “polycrystalline” here mean? Does it mean the whole sample was polycrystalline or a Cu nanoparticle is polycrystalline? A powder sample of single-crystalline Cu nanoparticles will also generate diffractions of different facets since the nanoparticles are not oriented through any self-assemble process.

Response:

Here, the polycrystalline is compared to the single-crystal Cu samples, which has only one lattice orientation in the sample, such as the sample in Scientific reports, 2014, 4(1): 1-6. To avoid possible confusion, we delete this sentence.

3. XRD patterns indicate the existence of Cu₂O, but Cu⁺ is not considered in the XPS fitting in Figure 2C. Usually, large satellite features indicate the existence of Cu²⁺. Cu⁺ doesn't show strong satellite feature. The binding energies of 2p electron in Cu⁺ and Cu⁰ are similar, but Cu⁺ and Cu⁰ can be distinguished by Auger spectrum. Therefore, the Auger spectrum of the catalysts should also be analyzed.

Response:

Thanks for your suggestion. We measured the Auger spectrum of AgCu SAA, AgCu SANP and AgCu NP. As seen in revised Figure S4B in supporting information, also see below (Response Figure 1, new figure S4b), the Auger spectrum proves the existence of Cu⁺ in those samples, which agree well with the Cu₂O peak in XRD results. Related analysis has also been added into the revised main text in page 4 of the manuscript. It states “Auger spectrum further prove the existence of Cu⁺, as shown in Figure S4B, which agree well with the XRD results.”

Response Figure 1. Auger spectra of AgCu SAA, AgCu SANP and AgCu NP.

4. Figure S2G: “EDX of region 2, which only shows the signal of Cu without Ag due to the low content of Ag in AgCu SAA composition.” If Ag single atoms can not be detected by EDX, then why the EDX mapping in Figure S3E still shows the uniform distribution of Ag? An enlarged EDX spectrum of the AgCu SAA region need to be provided to show whether the signal of Ag is above the noise.

Response: EDX STEM can analyse the chemical composition of materials at low magnification. However, at the atomic scale, the complex electron scattering between the number of X rays detected and number of atoms the probe interacts with, making it impossible to directly relate x ray counts to the number or density of atoms (Microscopy and Microanalysis, 2016, 22:1432; Microscopy and Microanalysis, 2017, 23(3): 513-517). We can only use the low mag EDX to examine the existence of this element and overall redistribution in figure S3. But it cannot be used at atomic scale to interpret. There is no Ag signal detected in this localized region in Figure S2G, even in an enlarged EDX spectrum. It is just below the detection limit at local region. It does not mean Ag atom does not exist in the local region. We added a few sentences at the page 3 of the manuscript. It states “However, at the atomic scale, the complex electron scattering between the number of X-rays detected and number of atoms the electron probe interacts with, make it impossible to directly relate X-ray counts to the number of density of atoms.^{22,23} Thus, we cannot confirm the existence of Ag single atoms by point spectrum (Figure 2G). In order to confirm the existence of Ag single atoms, atomic scale imaging is required.”

5. In the EXAFS fitting of AgCu SANP, the Ag-Ag and Ag-Cu coordination number are 7 and 1, respectively. Indicating more Ag atoms were in Ag NPs instead of SAA. The Cu-Cu (metal + oxide) and Cu-Ag coordination number are 3.5 and 1, respectively. Considering that the fraction of Ag was only 1 wt% in this sample and only a small fraction of Ag atoms was SAA, the ratio of coordination numbers of Cu-Ag/Cu-Cu as 1:3.5 seems not very reasonable. The coordination number of Cu-Ag seems too high.

Response:

Thanks for the reviewer’s comments on the EXAFS fitting results. EXAFS measures the average structure of every possible structural phase in the material. Based on the Ag single atom sites, Ag would bond directly with Cu; so those single atomic Ag signals would only average with Ag nanoparticles signal, which is more critical to represent the proportion of Ag single atom and Ag nanoparticles. No other Ag-containing structure form involved. However, the Cu structure is much more complex than Ag, which contains CuO, Cu₂O, and Cu atoms in the Cu cluster directly bonded to Ag, and the Cu atom in the Cu cluster indirectly bonding to Ag. Based on the Ag fitting and DFT modeling, only a small

amount of Cu bonding with Ag in the Cu cluster. Those complex mixtures of Cu phases and a small amount of Cu-Ag bonding make the ratio estimation from Cu EXAFS is not less certain or reliable. As shown in the following response figures 2-3 and response table 1, we could even fit the Cu EXAFS without any contribution from Ag-Cu, but the lack of Ag-Cu contribution in the Ag EXAFS cannot reproduce the spectrum well. In our manuscript, therefore, we added an Ag-Cu scattering path in Cu EXAFS to make consistence and demonstrate that we could also see it from Cu K-edge. To make clearer the interpretation of the coordination number, we add the following comments in the EXAFS fitting table S1 in the supporting information: “Due to complex mixture phases of Cu structures, Cu₂O, CuO, Cu atoms bonded to Ag in Cu cluster, and Cu atoms not bonded to Ag in Cu cluster, the coordination number of Cu-Ag extracted from Cu EXAFS spectral is just the demonstration of existence Cu-Ag.”

Response Figure 2: Fourier Transform R-space of Cu K-edge EXAFS experimental and fitting spectrum of AgCu SANP without Cu-Ag scattering path

Response Figure 3: Fourier Transform k-space of Cu K-edge EXAFS experimental and fitting spectrum of AgCu SANP without Cu-Ag scattering path

Response Table 1 Fitted EXAFS parameters at the Cu and Ag K-edge for AgCu SANP without Ag-Cu scattering path.

AgCu SANP	CN	error	R(Å)	error	E ₀ (eV)	σ ² (Å ²)	R-factor
Cu-O	2.20	0.48	1.955	0.018	8(2)	0.0023(1)	0.0212
Cu-O	1.10	0.24	2.683	0.087		0.0070(1)	
Cu-Cu(oxide)	2.20	0.48	3.062	0.043		0.0118(1)	
Cu-Cu(metal)	1.29	0.52	2.556	0.018		0.0008(5)	

6. Line 97-98: “The positive shift of the adsorption edge of AgCu SANP in the Cu K-edge also demonstrates the electron loss of Cu (Figure S5).” The XRD has shown the existence of CuO and Cu₂O in this sample. The edge positions of CuO and Cu₂O are more positive than Cu foil. Therefore, the positive shift of the edge did not indicate the electron transfer from Cu to Ag. The standard spectra of CuO and Cu₂O should be added in Figure S5A. If possible, the author should reduce the sample by H₂ at relative low temperature to remove the oxides and keep the Cu-Ag distribution, then collect the XAS spectra, to exclude the interference from oxide.

Response:

Thanks for pointing out the inaccurate argument here. The positive shift of the adsorption edge of AgCu SANP in the Cu K-edge may be also caused by the existence of Cu₂O and CuO in our system. However, the edge shift of Ag XANES and no Ag-O scattering path existing in Ag EXAFS confirms that Ag and Cu would form a bonding to allow the exchange of some electrons between them. We revised the statement in page 4 of our manuscript. We also added the standard spectra of CuO and Cu₂O in figure S5 in the supporting information for comparison.

Revised: “As shown in the Ag K-edge X-ray absorption near-edge structure (XANES) spectral (Figure 2D), the adsorption edge (E₀) around 0.5 of AgCu SANP shows a slight shift to lower energy compared to that of Ag foil, indicating electron transfer from Cu to Ag due to the formation of Ag-Cu bond. **Since the co-existence of Cu₂O and CuO, it is hard to distinguish either the positive shift of the adsorption edge of AgCu SANP in the Cu K-edge is caused by electron transfer from Cu to Ag or the electron transfer from Cu to O (Figure S5).** However, Fourier transform (FT) of the k²-weighted extended X-ray absorption fine structure (EXAFS) curve of the Ag K-edge of AgCu SANP shows both

Ag-Ag and Ag-Cu coordination bond (**Figure 2E**), and lack of Ag-O scattering path, which confirms the electron transfer between Cu and Ag.”

Response Figure 4: New Figure S5a

Due to the limited experimental conditions, it's very hard for us to use H₂ reducing the samples at synchrotron. We are so sorry on this point and hope you can understand.

7. In the EXAFS fitting of Cu in AgCu SANP, the bond length of first Cu-O shell was 1.952 Å, longer than the Cu-O bond in CuO (1.947 Å) and Cu₂O (1.849). What is the possible reason? The existence of Cu₂O was not considered in the fitting.

Response:

For the theoretical model CuO and Cu₂O we used for EXAFS fitting, the Cu-O in CuO is about 1.956 Å and 1.848 in Cu₂O. Our fitting bonding length is about 1.952 Å. Considering the error bar, the value falls between two values. For EXAFS fitting, it is a local structure instead of the entire bulk structure or crystal structure for XRD. We only need Cu-O scattering path either from Cu₂O or CuO, and the simulation process would adjust the scattering path length (bonding length) to fit the real experimental data. In addition, the real system would be different from the idea model which may have some cell expansion or distortion, particularly in the nanoparticle or nanocluster system. It could also increase the bonding length to some extents.

8. Line 125-126, “As presented in Figure 3E, AgCu SAA presents a much higher FE than that of Cu NP, strongly indicating the improvement of C-C coupling kinetics by Ag single-atom in Cu lattice.” The FE of Cu NP is about 87%, and that of AgCu SAA is around 100%. This only indicate HER was suppressed on AgCu SAA. To show the kinetics on two catalysts, you should compare

the reaction rate, namely partial current density, rather than FE of C₂₊ products. Similarly, plots of partial current density should be provided for CO₂ reduction. Since the total current densities on Cu and AgCu SANP are similar (Figure 3C), the enhancement factor of partial current density of C₂₊ on AgCu ASNP seems less than 2 compared with Cu. The difference between AgCu SAA and AgCu ASNP is even smaller. The enhancement factor less than 2 is not very significant in kinetics. The increasing of FE of C₂₊ is more related to the suppression of formation of other minor products.

Response:

Thanks for pointing out this and sorry for our misunderstanding on the kinetics. We agree with you that the increasing of FE of C₂₊ is more related to the C-C coupling selectivity rather than the kinetics. Actually, the main novelty of this work is the high selectivity of C₂ products, which is agree well with the C-C coupling selectivity improvement. We have corrected the statements based on your comments and also provide partial current density in new figure S11. This suggestion is very helpful.

We revised the statement in page 5. It states “Again, the concept of this proposed AgCu SANP cascade catalysis is that the Ag single-atom on Cu can promote the C-C coupling selectivity, while the Ag nanoparticles can produce local CO from CO₂. To further prove this point, CO reduction experiments were carried out to study the C-C coupling selectivity. As presented in Figure 3E, AgCu SAA presents a much higher FE than that of Cu NP, strongly indicating the suppression of HER and improvement of C-C coupling selectivity by Ag single-atom in Cu lattice.”

“Similarly, **Figure S11** shows the C₂₊ partial current density at -0.65V increased from 353 mA cm⁻² (Cu NP), 553 mA cm⁻² (AgCu SAA) to 677 mA cm⁻² (AgCu SANP), which proves that Ag increases the local CO concentration and expedites the C-C coupling.

Response figure S5: Figure S11. (A) The partial current density of C₂₊ products of Cu-based samples; (B) The FE values of various products during the long-term stability test. Noted that the FE of H₂ products from competitive hydrogen evolution reaction is not presented.

9. Following last comment, to compare the intrinsic kinetics of C₂₊ formation on different catalysts, maybe it's better to conduct the CO₂ reduction tests in H cell with low loading of catalyst, to guarantee all catalyst particles are accessible for CO₂ in solution. The current density needs to be kept low to avoid the rate being limited by mass transport. Then by comparing the ECSA normalized partial current density of C₂₊ products, the intrinsic activity of different catalysts can be compared. Under this situation, the FE of C₂₊ may be much lower than the value in flow cell, but more kinetic insights can be obtained.

Response:

Thanks for your recommendation on the kinetics tests. Our study is conducted in a flow basic electrolyte (1M KOH) in the flow cell. But if we move the experiments in a H-cell, the electrolyte inevitably change into neutral solution, because CO₂ will react with KOH and form KHCO₃ or K₂CO₃. That means, there is no possible way to keep the electrolyte in H-cell to be the same with that in flow-cell. If electrolytes change, pH will also be different, and the experiment results in H-cell may be totally different to that in flow-cell. We tried to simulate the flow cell in a H-cell by importing CO₂ gas into KOH electrolyte, but it's not stable. Thus, we conducted the test in H-cell with a neutral electrolyte KHCO₃ electrolyte. We tried to test in neutral electrolyte (KHCO₃), but the activity trends changed a lot. The pH of electrolyte also affects the performance a lot, which agree well with previous work, such as ACS Energy Letters, 2020, 5(10): 3101-3107; ACS Energy Letters, 2018, 3(4): 812-817; etc. It is very hard to study the kinetics of CO₂ reduction under alkaline electrolyte. Considering the emphasis of this work is on the selectivity of C₂ products, we think the lack of kinetics doesn't affect the main conclusion.

To further demonstrate our understanding, we conducted the H-cell test using 0.1M KHCO₃ electrolyte using glassy carbon electrode with low loading of catalyst. As shown in below images, the CO₂ reduction activity in neutral electrolytes is relatively low. In particular, the highest total CO₂ reduction FE are lower than 30% with a highest C₂₊ FE of ~23% in AgCu SANP. Especially, the major C₂₊ product over AgCu SANP is acetate, which is much different to that in KOH electrolyte.

Response Figure 6 H-cell measurement in KHCO₃ neutral electrolyte using glassy carbon with low catalyst loading. (A-D) Faradic efficiency (FE) results of Cu NP, AgCu SAA, AgCu SANP, and Ag NP under different potentials in 0.1M KHCO₃.

We further measured the ECSA of all catalysts and normalized the current density by ECSA measured in the KHCO₃ neutral condition in H-cell. As seen in below figure, the maximum ECSA normalized partial current density of C₂₊ products of AgCu SANP are still higher than other catalysts. What should be noted that the C₂₊ products composition are different that in KOH electrolyte. This results probably indicates that the AgCu SANP has best C-C coupling ability. But we still doubt this conclusion can be applied into the KOH electrolyte due to the difference in electrolyte.

Response Figure 7. ECSA measurement in H-cell in KHCO_3 neutral electrolyte. (A) ECSA; (B) Maximum ECSA normalized partial current density of C_2^+ products; (C-F) ECSA normalized current density of various CO_2 reduction products of Cu NP, AgCu SAA, AgCu SANP, and Ag NP under different potentials in 0.1M KHCO_3 .

10. Line 152-153, “To model AgCu SAA, we adapt an Ag-doped Cu model structure.” What is the structure of this “Ag-doped Cu model”? Line 167-168, “In pure Cu(100) surface the Cu-Cu

bond length is 2.57 Å, however, after doping, this is compressed to 2.50 Å at the surface.” Which Cu-Cu bond is compressed? The bond between two Cu next to Ag atom? Please draw the structure of cells used for simulation and indicate the bond mentioned.

Response:

We thank the reviewer for this comment. As shown below, in the pure Cu(100) surface the distance between Cu-Cu bond is 2.57 Å, and in the Ag-doped Cu model the distance between neighbouring Cu-Cu atoms become 2.50 Å. The reduced distance between Cu-Cu atoms adjacent to Ag atom in the Ag-doped Cu (100) structure drives the reaction in the favourable direction.

We revised the statement in page 6-7 in the manuscript. It states “The Cu-Cu bond length is 2.57 Å on pure Cu(100) surface, Cu-Cu bonds next to the Ag site are compressed to 2.50 Å after doping (**Figure S14**). The compressive strain creates asymmetrical Cu sites next to the Ag atom (i.e., first-neighbour atoms labelled ‘B’, **Figure S15**), which provide active sites for the C-C coupling.”

Response Figure 8. Cu-Cu bond distance comparison before Ag doping (A) and after Ag doping (B).

11. Line 161-163: “Prior studies have reported that on Cu electrodes, the C-C coupling can occur through *CO-*CO dimerization, *CO \rightarrow *CHO, or *CO-*COH with *CO-*CO coupling being less feasible with a higher transition state (TS) energy (> 1 eV).31-33” Do the authors mean *CO-*CHO and *CO-*COH couplings are less feasible than *CO-*CO coupling? If so, why the author only simulated *CO-*CHO and *CO-*COH couplings in the following sections. I think more previous works regard *CO-*CO coupling as the predominant pathway for C-C bond formation, including references 31-33. In some reports, *CO-*CO is an electron-transfer step decoupled with proton transfer. Decoupled electron-proton transfer is more difficult to treat in DFT simulation. Is this part of the reason the authors chose to simulate *CO \rightarrow *CHO and *CO-*COH couplings.

Response:

We thank the reviewer for this comment and would like to mention that in the original version of our manuscript in line 161-163, we wrote the C-C coupling can take place via *CO-*CO, *CO-*CHO, or

*CO-*COH coupling. However, the C-C coupling via *CO-*CO coupling is less feasible than those through *CO-*CHO and *CO-*COH. This has been demonstrated by various reports that the transition state energy of *CO-*CO coupling is 1 eV higher than *CO-*CHO and *CO-*COH (Ref. 31, 32). For this reason, we originally focused our study on calculating the *CO-*CHO and *CO-*COH coupling. We also agree with the reviewer that the decoupled electron-proton transfer in *CO-*CO coupling is more difficult to treat in DFT simulation. As a matter of fact, our DFT calculations for the *CO-*CO dimerization were revealed to be hard to converge.

12. Figure 4a and 4b only show the energy diagram on Ag-doped Cu (100). The authors should overlay the energy diagram on pure Cu(100) to show how much difference the Ag atoms induced. Line 170-174: “As shown in Figure 4A, the TS values of C-C coupling via *CO-*CHO and *CO-*COH reactions are 0.51 and 1.10 eV, respectively, ... A transition state energy of 0.55 eV barrier is obtained, which is slightly higher than the Ag-doped Cu(100) surface.” What is the common error for DFT simulated activation energy? Is 40 meV difference of couple barrier a significant difference? Does the doping of Ag significantly facilitate the coupling step?

Response:

We thank the reviewer for drawing our attention to this important point and included the energy diagram with pure Cu(100) surface in response Figure 9 (new Figure 4 in manuscript), and updated the caption, which now better shows the impact of Ag doping on the transitions states.

Response Figure 9. Mechanistic studies by the density functional theory calculations. (A) Comparison of the C-C coupling activation barrier for the *CHO (on CuAg and pure Cu) and *COH intermediates (CuAg); (B) the lowest free energy pathway for the formation of ethanol (orange), ethylene (blue) and acetic acid (magenta) on CuAg and (C) the associated chemical formula for each elementary step in (B).

Regarding the DFT error, the reviewer is right. The DFT error in estimating the bond energies using PBE is 0.2 eV as reported by Wellendorff, et al., *Surface Science*, **640**, 36 (2015). This error is higher than the energy difference of 40 meV obtained in our calculations. We would like to mention that we revisited our calculations and found a better transition state in our calculations that places the energy difference between transition states in pure and Ag-doped copper at 0.1 eV. But even then, this is still in the range of DFT error and we cannot fully conclude that the Ag-doping has a positive impact on the transition state energy of *CO - *CHO when compared to pure Cu surface. However, the key point we are trying to make in this study is that in the presence of Ag the local concentration of CO near the surface is increased due to CO_2RR to CO on Ag sites. This facilitates the coupling of CO on the adjacent Cu sites and these sites have a slight advantage in C-C coupling over pure copper. Thus, the synergistic effect between the two phenomena places this catalyst in a better position than pure copper.

To address the reviewer's concern, we included the above discussion, updated our results and revised our manuscript (highlighted in yellow) in page 7.

13. Line 174-175: "The main difference between the selectivity of Cu(100) and Ag-doped Cu(100) thus comes from the fact that Ag is deemed to produce CO from CO₂." Does "Ag" here mean Ag atom or Ag particle? Is Ag single-atom in Cu or Ag NP the catalyst for CO₂-to-CO? If Ag single-atom can play this role, then why AgCu SANP showed higher selectivity to C₂₊ than AgCu SAA in CO₂ reduction. Maybe, the authors should simulate the energy diagram of CO₂-to-CO on Ag single-atom and Ag metal to show which one is more active for this step and whether Ag NPs are necessary for the tandem reaction.

Response:

We thank the reviewer for this comment and agree that the terminology in our manuscript can be confusing. The Ag-doped Cu(100) refers to the Cu(100) surface doped with a single Ag atom (System 'b' in Figure S14 of the revised manuscript), which is meant to model the AgCu SAA, and the chemical environment surrounding the single Ag atoms in AgCu SANP.

As discussed in the manuscript, AgCu SANP shows higher selectivity toward C₂₊ products because of the cascade effect of Ag nanoparticles and single Ag atoms at the surface AgCu SANP. The nanoparticles in AgCu SANP trigger the reduction of CO₂ to CO, increasing the local concentration of *CO at the surface. On the other hand, the single Ag atoms in AgCu SANP facilitate the production of C₂₊ products via C-C coupling of *CO and *CHO on Cu sites near to Ag atoms (sketched in Figure 3A). The CO₂-to-CO reaction with AgCu SAA is reduced compared to AgCu SANP and the concentration of *CO is lowered. As a result, the C-C coupling is also reduced when compared to AgCu SANP.

Regarding the energy diagram of CO₂-to-CO on Ag single-atom and pure Ag surface, we performed additional calculations, and the results are summarized in the free energy diagram below (response figure 7, new figure S16). As can be seen, both pure Ag and Ag-Cu are active for CO₂RR to CO with a slight advantage of Ag-doped Cu over pure Ag. The new analysis is now included in the revised manuscript and a new plot (Figure S16) is included in the supporting information.

We added more discussion in page 7 of the manuscript. It states "On the other hand, Ag is known to produce CO from CO₂ (*Adv. Powder Mater.* **1**, 100012, doi:10.1016/j.apmate.2021.10.003 (2022)). The Ag sites both in the single atom (AgCu SAA) and nanoparticle (AgCu SANP) could convert CO₂-to-CO, however, higher formation of CO is expected with the latter because of the number of active sites. To assess the formation of CO on a single Ag atom and Ag nanoparticle, we compare the CO₂ reduction to CO on Ag-doped Cu and Ag(100) surface (**Figure S16**). The result shows that the Ag-doped Cu is a

better catalyst for CO₂ to CO reduction than the pure Ag surface (**Figure S16**). However, nanoparticles have more undercoordinated Ag sites (corners and kinks) than the lower Miller index surfaces. Those undercoordinated Ag atoms are the most active for CO₂ to CO reduction reaction (*ACS Nano* **15**, 7682-7693). The AgCu SANP catalysts are expected to have kinked and corner sites. Therefore, it is likely that the CO₂ to CO reduction takes place on both single Ag atoms and Ag nanoparticles.”

Response Figure 10. Figure S16. CO₂-to-CO reduction on Ag-doped Cu (black line) and Ag(100) (orange line).

14. Line 211: The Chemical section did not provide the information of Cu NPs and AgNO₃, the most two important chemicals used in this work.

Response:

Cu NPs are 100 nm Cu nanoparticles purchased from Sigma. AgNO₃ with a purity of 99% is also purchased from Sigma. That information was in the middle of the chemicals section. Considering its importance, we put that sentence at the front of that paragraph in page 9 of the manuscript.

Reviewer #2 (Remarks to the Author):

The author designed an AgCu single-atom alloy and Ag nanoparticle cascade catalyst for CO₂ to C₂₊ products conversion. The catalyst demonstrates 94% total C₂₊ products selectivity under 720 mA cm⁻² current density. The performance ranks at the top level of all the published results. Moreover, the author used a simple and reproducible synthesis method. In the meanwhile, the characterization is comprehensive and detailed. Overall, I support the publication of this manuscript if the following concerns are addressed.

Response:

Thanks for your encouraging comments on our work. Please see below for detailed response to your concerns.

1. When the AgCu SANP significantly improves the C₂₊ FE compared to Cu NP and AgCu SAA, the current density remains in a similar range. If the Ag increases the local CO concentration and expedites the C-C coupling, the improvement in current density should be more obvious. Could the author explain more about that?

Response:

Figure 3C presents the total current density of those catalysts at -0.65V, which consists of current density from C₂₊ products, C₁ products and competitive hydrogen evolution reaction (HER). Under your considered condition, if the Ag increases the local CO concentration and expedites the C-C coupling, the current density from C₂₊ products should be increased. To better compare the change of C₂₊ partial current density, we added a figure as new **Figure S11** in revised supplementary information, also see below as response figure 11. As seen in Figure S11, the C₂₊ partial current density at -0.65V increased from 353 mA cm⁻² (Cu NP), 553 mA cm⁻² (AgCu SAA) to 677 mA cm⁻² (AgCu SANP). Too much Ag in AgCu NP cause a rapid decrease of C₂₊ partial current density (170 mA cm⁻²) due to lack of enough Cu sites.

Response Figure 11. Partial current density of C₂₊ products of Cu-based samples.

2. Could the author explain why there is a 20% missing for the total FE of Cu NP in Figure 3B? It is not a fair comparison to say AgCu SAA improves C₂₊ FE if the Cu NP result has a large error. Similarly, the AgCu NP data in Figure 3B and AgCu SANP data in Figure 3E all miss a significant part of the total FE.

Response:

Those missed FE is mainly from the competitive HER. To avoid misunderstanding, we add an explanation in the caption of figure 3. Unfortunately, the GC in our lab is unable to detect H₂ together with all those CO₂ reduction products at the same time, so we have no H₂ data. However, all those CO₂ reduction products were measured with 3 repeated experiments and error bar are presented. We believe those results are reliable. We hope you can understand that.

3. Figure 3F uses "second" as the unit for the time scale, which is not reading-friendly. The author should use "hour" as a unit.

Response:

Thanks for your advice. We have changed the unit and revised Figure 3F. Please also see below as response figure 12.

Response Figure 12. Long-term stability results of AgCu SANP toward CO₂RR.

4. The author claims that ethanol formation is favored over ethylene. Then, why do Cu NP and AgCu SAA generate more ethylene than ethanol during CO₂RR?

Response:

We thank the reviewer for the question. We have performed the DFT calculations for the formation of C₂ products, i.e., ethanol, ethylene and acetate, from CO₂. In our calculations, we have found that the difference in free energy in the uphill steps of ethanol and ethylene (CH₂CH₂OH* → CH₃CH₂OH and CH₂CHO* → CH₂CH₂O, respectively, Figure 4B) is around 0.04 eV, which is within the DFT error (0.2 eV). Based on our DFT results, we cannot conclude that ethanol is favoured over ethylene or vice-versa. Instead, we believe that both ethanol and ethylene can be major products from CO₂RR on Ag-doped Cu(100). As well, we would like to note that our calculations do not take into consideration the solvent, pH and double-layer electric field effects. These can affect the stabilization of the intermediates and, consequently, the selectivity toward certain products (Tang *et al.*, *J. Phys. Chem. C* 2021, **125**, 26437; Liu *et al.*, *Nat. Commun.* 2019, **10**, 32).

We corrected the discussion in the revised manuscript, highlighted in yellow in page 8 of the manuscript. It states “Comparing ethanol and ethylene formation from CO₂ (which both are 12-electron reduction products), free energy diagram analysis revealed that the uphill steps after the coupling for ethanol and ethylene (CH₂CH₂OH* → CH₃CH₂OH and CH₂CHO* → CH₂CH₂O) are 0.4 and 0.44 eV, respectively. This difference of 0.04 eV is negligible (within the DFT error), which indicates that both ethanol and ethylene are equally probable products of CO₂RR on Ag-doped Cu. We would like to note that the solvent, pH and double-layer electric field effect were not included in the calculations, which could affect the stabilization of different intermediates and, consequently the selectivity of the final products.”

Reviewer #3 (Remarks to the Author):

In this paper, the authors disclosed the synergy effect between AgCu SAA and Ag nano particles for efficient CO₂RR. Although a high FE of 94% C₂+ products was achieved on their AgCu SANP, no enough evidence could support their results and mechanism.

Response:

Thank you for your valuable comments. We have answered all questions from you and revised the manuscript based on your comments. We hope those answer and revision can address those issues you raised.

1. Although a special structure was claimed, the traditional tandem reaction mechanism was still used to explain their results. How is the meaning to constructure this new structure?

Response: Tandem reaction mechanism is a classic but valuable mechanism in electrocatalytic CO₂ reaction. Many breakthroughs and new findings have been achieved based on this mechanism, such as Nature Catalysis volume 5, pages202–211 (2022); Adv. Funct. Mater. 2021, 31, 2101255; Nat Catal 3, 75–82 (2020); etc. This work intends to develop a highly selective electrocatalyst for C₂+ products, rather than developing a new reaction mechanism. Herein, we report a record high Faradaic efficiency (FE) of 94% toward multicarbon products over the as-prepared AgCu SANP, under ~720 mA cm⁻² working current density in a flow cell. We think the meaning of this work is the promising performance based on the new structure of the catalyts.

2. With the increase of the Ag, there is no doubt to constructure Ag nanoparticles. How about the proportion of Ag SA and nanoparticles? What is the main contribution to the reaction? How to determine Ag SA is close to the Ag nanoparticles? How many Ag SA will improve the C-C coupling and what is the case of nanoparticles close to each other? What is the effect of Ag SA far from the Ag nanoparticles?

Response:

a. How about the proportion of Ag SA and nanoparticles?

The proportion of Ag SA and nanoparticles was estimated based on the ratio of Ag-Cu bonds to Ag-Ag bonds, which was around 1:7 according to the extended X-ray absorption fine structure (EXAFS) fitting results (Figure S6 and Table S1).

b. What is the main contribution to the reaction?

Our hypothesis is that AgCu SAA serves as a C-C coupling site while Ag NP produce CO locally, as schemed in Figure 3A. DFT study investigates the reaction mechanism and demonstrate the reasonableness of our hypothesis.

c. How to determine Ag SA is close to the Ag nanoparticles?

According to the mapping images in Figure 1E, all elements are uniformly dispersed in AgCu SANP, which means atomic Ag atoms are also uniformly dispersed. So, no matter where the Ag NP locates in AgCu SANP, it should be closer to some Ag SA.

d. How many Ag SA will improve the C-C coupling and what is the case of nanoparticles close to each other?

To be honest, it's very hard to determine the detailed number of Ag SA which will improve the C-C coupling. What we can do is to adjust the ratio of Ag SA and Ag NP experimentally, and then conclude a best ratio based on the electrochemical performance results. In principle, those Ag NP close to each other will produce C_1 products, such as CO, etc. And this maybe the reason why AgCu SANP can't achieve 100% FE for C_{2+} products. In other words, we can only experimentally adjust the ratio of Ag SA and Ag NP to achieve a best structure and realizing a good performance.

e. What is the effect of Ag SA far from the Ag nanoparticles?

If we assume the Ag SA is far enough from the Ag NP, which means no CO supplement from Ag NP to Ag SA, the Ag SA would reduce CO_2 directly. That's the same case of AgCu SAA without Ag NPs, which can also produce C_{2+} products with lower FE. This may be the reason for the un-equal FE for C_{2+} products.

3. Usually, Ag shows weak adsorption to *CO. Why the authors considered it will increase the adsorption of *CO for Cu? It is wired that the adsorption of *CO could be enhanced by a weak adsorption one.

Response:

We think there is a misunderstanding here. It is correct that the adsorption of *CO on Ag is weak. Our DFT calculations also confirmed that the *CO molecule weakly adsorbs on the Ag atom. The adsorbed *CO on Ag-Cu bridge site is also weak and diffuses on the top site of the Cu atom (Figure S14 and Table S3), which is a more favourable site for *CO. The point we are trying to make is the larger atomic radius of Ag creates a surface strain on the Cu atom next to the Ag atom, with a Cu-Cu bond length of 2.5 Å, which is 0.07 Å shorter than the typical 2.57 Å of Cu-Cu bond on pure Cu(100) surface. As we pointed out in the original version of our manuscript, this strain creates asymmetrical Cu sites next to the Ag atom, which provides active Cu sites for the *CO adsorption and, hence, facilitates the C-C coupling.

To address the reviewer's comment, we have revised our discussion in page 6-7 in the manuscript to provide more clarity and included the following figure in the Supplementary Information of the revised manuscript (Figure S14). Please also see the figure as response figure 13 below.

Response Figure 13. Cu-Cu bond distance comparison before Ag doping (A) and after Ag doping (B).

4. From the CO₂RR, the authors just take the very beginning 100s or less data. In the early stage of the reaction, there are always some fake data. Thus, data from over 20 min reaction is recommended to show its performance.

Response:

Thanks for point out this. As seen in Figure S10 and S12, all the current are stable after about 100-200s. And we start to collect the GC data when the current is stable. As for the liquid products, we collect and measured the electrolyte after the whole 600s (10 minutes) test. All the tests are repeated three times and relative error bars are calculated and presented. In addition, the FE the C₂₊ FE only decreases a little from 98% to 88% during the 13h test (total working time 9.75h), as shown in Figure 3F. The reported performance is reliable and reproducible.

5. In the CO reduction, the AgCu SANP showed much lower ethanol generation than that of AgCu SAA and Cu NPs. While, it showed much higher ethanol generation than those of AgCu SAA and Cu NPs in CO₂RR. This is opposite for their explanation. The same problems are similar for ethylene and acetic acid.

Response: We thank the reviewer for pointing out this question. The DFT part of this work is mainly focused on the CO₂RR rather than CORR. With respect to CO₂RR computational results match the experimental results for CO₂RR Figure 3B. In this study, we showed the CO₂-to-CO reaction is promoted by the Ag nanoparticles in AgCu SANP whereas the C-C coupling occurs at the Cu sites-facilitated by single Ag atoms in the sample. When the sample is fed with CO₂ gas, both reactions can take place in a cascade fashion, as detailed in the manuscript.

Considering it's hard to separately study the *CO coupling step in CO₂RR, CORR test was measured as a model reaction to experimentally demonstrate and compare the C-C coupling ability of those samples. Figure 3E shows that most of the products are C₂ products with extremely small C₃ products, indicating that the main reaction in CORR is C-C coupling and is suitable for simulating the *CO coupling reaction in CO₂RR. Here, the CORR experiments shows that AgCu SAA exhibited highest FE

of C₂₊ products among the samples. This result confirms the C-C coupling selectivity of the Ag single atom site on Cu, and further indicating the reasonableness of the cascade catalysis concept on the C₂₊ production.

What should be noted that, the CORR experiments is not the same with the *CO coupling reaction in the CO₂RR. Therefore, there is no evidence to prove that the results of CORR should be the same trend with CO₂RR. Here we only using the CORR to compare the C-C coupling ability of those samples, rather than to simulate the CO₂RR products.

6. The DFT simulation is too simple.

Response:

The DFT calculations is used to study the catalytic active sites and reaction mechanisms, which demonstrates the cascade catalysis concept in this work. We are not sure, in what sense our DFT calculations seem simple to the reviewer. It would be great if the reviewer can clarify this point.

7. There many typing error, such as Figure S7, S8 were wrong in the experimental part.

Response:

Thanks for pointing out this. We have revised the figure number in the experimental part and checked the main text again.

REVIEWERS' COMMENTS

Reviewer #1 (Remarks to the Author):

The authors have addressed most of the issues in my comments. The quality of this manuscript has been improved. I understand that the clear characterization of a catalyst containing Ag single atom and Ag NPs is difficult, and the intrinsic activity of this kind of material is difficult to measure. Anyway, this work provides a very selective catalyst for CO₂ reduction to C₂+ products, and the hypotheses and characterizations in the present manuscript is self-consistent. I think this manuscript can be accepted.

Reviewer #2 (Remarks to the Author):

I am satisfied with the revised version and, therefore, fully support the publication of this paper.